# Chronically Increased Levels of Circulating Insulin Secondary to Insulin Resistance: A Silent Killer

**DOI:** 10.3390/biomedicines12102416

**Published:** 2024-10-21

**Authors:** Serafino Fazio, Paolo Bellavite, Flora Affuso

**Affiliations:** 1School of Medicine, Federico II University, 80100 Naples, Italy; 2Homeopatic Medical School of Verona, 37121 Verona, Italy; paolo.bellavite@gmail.com; 3Independent Researcher, 80100 Naples, Italy; flora.affuso70@gmail.com

**Keywords:** insulin resistance, hyperinsulinemia, type 2 diabetes, treatments, adverse effects, risk factor, cardiovascular disease, cellular senescence, cancer, neurodegenerative disease

## Abstract

Despite all the progress made by science in the prevention and treatment of cardiovascular diseases and cancers, these are still the main reasons for hospitalizations and death in the Western world. Among the possible causes of this situation, disorders related to hyperinsulinemia and insulin resistance (Hyperin/IR) are still little-known topics. An analysis of the literature shows that this condition is a multiple risk factor for type 2 diabetes, cardiovascular diseases, cellular senescence and cancer, and neurodegenerative diseases. Hyperin/IR is progressively increasing worldwide, and its prevalence has now exceeded 50% of the general population and in overweight children. Asymptomatic or poorly symptomatic, it can last for many years before manifesting itself as diabetes, cardiovascular disease, neoplasm, cognitive deficit, or dementia, therefore leading to enormous social and healthcare costs. For these reasons, a screening plan for this pathology should be implemented for the purpose of identifying people with Hyperin/IR and promptly starting them on preventive treatment.

## 1. Background

Notwithstanding the notable and continuous scientific advances made in the prevention and treatment of cardiovascular pathologies and tumors, they are currently the main causes of hospital admissions and deaths [1]. Furthermore, the aging of the population appears to be associated with a notable increase in neurodegenerative diseases. This entails significant social and healthcare costs.

Increased levels of circulating insulin (here referred to as Hyperin) secondary to insulin resistance (IR) represent a condition that predisposes to many diseases. This determines a silent pandemic, which produces enormous healthcare costs and an increase in hospitalizations and deaths.

Unfortunately, Hyperin/IR is a condition rapidly growing worldwide, and its prevalence has now exceeded 50% in the general population and is continuously growing [2,3]. In a recent study in Latin America, the prevalence of insulin resistance among overweight children and children with obesity was 57% and 72%, respectively [4].

This disorder has been commonly associated with the onset of type 2 diabetes, but its high prevalence and the multiple metabolic and cellular actions of insulin prompt consideration of its possible consequences in other important fields of clinical medicine.

This review of the literature on the topic aims to demonstrate how Hyperin associated with IR, if neglected, can cause serious and often irreversible damage to our body. The manuscripts used for the review were searched on medical literature databases such as PubMed, Scopus, Web of Science, etc., using the following keywords: Insulin resistance, hyperinsulinemia, type 2 diabetes, treatments, adverse effects, risk factor, cardiovascular disease, cellular senescence, cancer, and neurodegenerative disease.

Only peer-reviewed manuscripts published in English and in extenso from the year 1980 to 2024 in well-regarded journals were used.

## 2. Definition and Causes of Insulin Resistance

IR can be defined as a condition in which a given quantity of insulin produces a reduced metabolic result, in terms of glycemic control, compared to what is expected [3,4,5]. For this reason, in order to maintain normal glycemic values, the pancreas of subjects affected by IR are forced to secrete greater quantities of insulin, with the result being that Hyperin is a constant and fundamental characteristic of IR. On the other hand, Hyperin contributes to the onset of IR through various molecular mechanisms, thus generating a vicious circle that is self-sustaining and tends to worsen.

The mechanisms through which IR can develop in our body and those linked to Hyperin damage of some organs are very complex and still not completely clarified. There may be a defect at the receptor or, more likely, post-receptor level. However, there are also some certainties. We know two of the pathways triggered by the binding of insulin to its receptors, their mechanisms, and what happens to these pathways at the post-receptor level in the case of IR. They are the phosphoinositide 3 kinase (PI3K) and the mitogen-activated protein kinase (MAPK) pathways.

Insulin determines its multiple metabolic and non-metabolic actions through binding to its transmembrane receptors located on the target cells of its action. The intracellular domain of the insulin receptor belongs to the tyrosine kinase receptor protein family. The interaction of insulin and its receptors determines the activation of numerous protein kinases and gene transcription factors. Among these, the most important and best known, as already mentioned, are the PI3K pathway, which mainly regulates metabolic effects and the secretion of nitric oxide (NO), and the MAPK pathway, which is instead responsible for gene expression effects, cell growth and differentiation, and the production of endothelin-1 (ET-1) at the vascular level [3,5,6].

In the presence of an IR condition, there is mostly a malfunction at the PI3K post-receptor level, which mainly regulates the metabolic actions and the formation of NO, while the functioning of MAPK is little or not at all altered so that the non-metabolic actions of insulin and, in particular, the stimulating action on cell proliferation and on the secretion of ET-1 are promoted due to the chronic effects of Hyperin. Therefore, the different behaviors of these two pathways resulting from IR determine, over time, very important alterations in the target organs of insulin [3,5].

There are many causes that can produce IR, although it is likely that not all of them are known. Visceral obesity is a very important risk factor for the development of IR that plays a crucial role in the pathogenesis of this condition. Increased visceral fat releases large quantities of free fatty acids into the circulation. They alter insulin signaling pathways in their main target organs. In the liver, they determine an increased production of glucose due to reduced storage of the latter in the form of glycogen, increased production of triglycerides, and low-density lipoprotein cholesterol (LDLc). The increase in triglycerides and LDLc and the reduction in HDL cholesterol, which is characteristic of the IR condition, presents a profoundly atherogenic profile [7,8]. Free fatty acids also reduce insulin sensitivity at the muscle level by inhibiting insulin-mediated glucose uptake. The increased level of free fatty acids and glucose also produces an increase in oxidative stress and the formation of advanced glycation end-products. In addition to the mechanism linked to the increased production of free fatty acids, adipose tissue increases the degree of IR determining a chronic pro-inflammatory state. Adipose tissue is an active endocrine–paracrine organ. In the presence of visceral adiposity, alterations in the effects of leptin occur, leading to vascular inflammation, increased oxidative stress, and the hypertrophy of vascular smooth muscle cells, alterations all predisposing to the development and progression of atherosclerosis. Furthermore, there is a reduction in the production of adiponectin, which is an anti-inflammatory peptide whose circulating levels are inversely correlated with the degree of IR [9,10,11].

There are certainly genetic causes, but they are the minority. Among these, type A IR syndrome is a rare genetic disorder characterized by severe IR. In affected women, the main features of the condition appear during adolescence. Many affected women do not start menstruating by age 16 (primary amenorrhea), or their cycles are scanty and irregular. They develop ovarian cysts and excessive growth of body hair (hirsutism). Acanthosis nigricans is also often present. Unlike most people with IR, these individuals are generally not overweight. The characteristics of type A IR syndrome are more subtle in males. Some of them have low blood sugar as the only initial sign, and others may also have acanthosis nigricans. In most cases, males with this condition seek medical attention only in adulthood, when diabetes appears. IR syndrome type A is estimated to affect 1 in 100,000 people worldwide. Because women have more evident health problems associated with this condition, it is diagnosed more often in women than in men [12]. 

Although it is not yet perfectly clear whether the IR appears first and then the hyperinsulinemia (as one might think) or vice versa, these two conditions are, in the vast majority of cases, chronically associated and are rapidly and constantly growing throughout the world. Furthermore, since it is asymptomatic or poorly symptomatic, it goes unrecognized for years, and constitutes what could be defined as a silent pandemic [3,5]. Over the centuries, there has been a notable change in our lifestyle, characterized by a progressive increase in caloric intake in favor of foods rich in carbohydrates and highly processed, with a simultaneous reduction in physical activity, and this begins in children and persists into adulthood. This notable change in lifestyle, also associated with the increased stress and competitiveness of modern life, which, as is known, stimulates the secretion of diabetogenic hormones (above all, cortisol and GH), has produced a progressive increase and spread of IR and Hyperin [13]. Some studies have shown that high-sodium diets impair insulin sensitivity, although the results are not entirely consistent [14]. Also, many drugs induce insulin resistance. Among the best known there is cortisol [15]. Growth hormone (GH) therapy antagonizes insulin’s action in the target tissues, consequently increasing glucose production from the skeletal muscle and liver, while decreasing glucose uptake from adipose tissue. Therefore, insulin secretion is increased to compensate for the increase in glycaemia after GH administration [16]. Furthermore, it has been shown that protease inhibitors can increase IR by altering GLUT-4, which is the most important transporter of glucose in the target cells, stimulated by the action of insulin [17]. Also, atypical antipsychotics can induce insulin resistance and postprandial hormonal dysregulation independently from weight gain [18].

A very recent review article highlighted that adipose tissue-derived extracellular vesicles represent a possible new mechanism of cross-talk between organs. These vesicles contain proteins, lipids, and nucleic acids that can modify the phenotype and function of the target organs of insulin action. These vesicles released from adipose tissue can lead to the development of IR, non-alcoholic fatty liver, and polycystic ovary syndrome. In the near future, this mechanism could become a therapeutic target for the treatment of these pathologies [19].

## 3. Diagnosis

The euglycemic–hyperinsulinemic clamp technique gives us the diagnostic certainty of IR, but it is not conceivable that it should be employed for mass screening, so it is used almost exclusively for scientific research purposes [20]. However, there are many surrogate indices of IR that can be used for this purpose. Although there are many that are all useful, at least three surrogate indices are simple to obtain and are reliable [Table 1].

The three indices include the homeostatic model assessment index (HOMA-IR), which is calculated through the simultaneous fasting measurement of glycemia and insulinemia, with a cut-off of 2.5 in adults and 3.6 in children; the triglyceride–glucose index (TyG), which is obtained by simultaneously measuring triglycerides and fasting blood sugar, with a cut-off value of 8; and the ratio between triglycerides and HDLc with a cut-off of 2.75 in men and 1.65 in women. These three surrogate indices of IR have also been shown to be excellent independent markers of adverse cardiovascular events [20,21,22].

Those mentioned in Table 1 are the simplest and most practical indices, but for a more complete diagnosis, there are also other interesting biomarkers that are related to IR/Hyperin, including respiratory metabolites to assess systemic metabolic dysregulation [23], serum levels of molecules potentially related to cognitive decline (βA42 and PSEN1) [24], adiponectin [25], and the inflammation marker NHR (Neutrophils to the HDL/Cholesterol Ratio Index) [26]. For an even more individualized prognosis and to guide therapeutic strategies, the study of the expression of IR biomarker genes in muscles is becoming increasingly interesting [27]. Finally, since the intestinal bacterial flora is fundamental in the development of insulin resistance [28], the study of the microbiota and microbiota-derived metabolites could be of great prognostic interest in monitoring prevention and therapeutic interventions.

## 4. Effects of Hyperinsulinemia Associated with Insulin Resistance

Excess insulin in the blood associated with IR has repercussions on many districts precisely because this hormone has pleiotropic metabolic and epigenetic effects.

### 4.1. Cardiovascular Effects

Hyperinsulinemia associated with IR determines an alteration of circulatory homeostasis defined as endothelial dysfunction. It is caused by the prevailing synthesis and secretion of ET-1, compared to a reduced availability of NO, by the cells of the wall of the arteries and arterioles [Figure 1]. This causes vasoconstriction with the reduction in circulatory flows to the tissues. Furthermore, hyperinsulinemia, acting as a growth stimulus, determines the increase in vascular thickness and parietal stiffening, a phenomenon that is prodromal of the development and worsening of hypertension and atherosclerosis [29,30]. Hyperin/IR, in addition to the alteration of glucose metabolism and the direct actions of insulin at the renal level, at the level of the sympathetic nervous system, and as a growth factor, also determines a triad of frankly pathological factors at the cardiovascular level, namely high triglyceride levels, low HDLc levels, and the appearance of small and dense lipoproteins which, together with the endothelial dysfunction already described, contribute to the formation and progression of atherosclerotic plaque [31,32].

In addition, in IR conditions, there is solid evidence of the fact that Hyperin is an important cause of arterial hypertension through the renal reabsorption of sodium and increased sympathetic tone [33,34]. Insulin receptors are located in the renal tubules, and it has been seen that their stimulation by insulin determines increased Na+ and water reabsorption.

In addition to this, a close relationship has been demonstrated between the increased levels of circulating insulin and the enhanced activity of the sympathetic nervous system. This chronic disorder can eventually lead to the concentric remodeling of the LV, a recognized predictor of heart failure with preserved ejection fraction (HFpEF) [Figure 2] [35,36,37].

Arterial hypertension was associated with both increased insulin levels and enhanced sympathetic activity, with a relationship demonstrated both in the whole population and after correction for BMI and body fat. Furthermore, it has been assessed that, by lowering insulin levels in obese subjects, decreases in plasma norepinephrine and blood pressure values can be obtained [33].

A fairly recent study, performed in 88 hypertensive Sub-Saharan African patients with myocardial hypertrophy, has shown that obesity and Hyperin/IR predicted the increase in left ventricular mass. Therefore, the authors suggested that could be particularly important to correct obesity and IR/Hyperin to counteract the development of LVH in these patients [35].

Another study was performed in Japan, where 210 normotensive subjects and 180 patients with mild or moderate hypertension were studied using echocardiography and measurements of glycemic metabolic parameters. The sum of glucose or Hb A1c levels in the whole group of subjects and the sum of insulin levels (or insulin values 2 h post-load) in non-diabetic subjects were highly related to the relative LV wall thickness values, independently of age, systolic blood pressure, and BMI. Therefore, the authors concluded that hyperglycemia and Hyperin could stimulate LV concentric remodeling in normotensive subjects and in patients with mild or moderate hypertension. In a study by our group performed using Doppler echocardiography, 59 patients with IR/Hyperin showed both increased LV mass and relative wall thickness, together with LV diastolic dysfunction. All these parameters improved following the treatment with an insulin-sensitizing substance [3,38].

### 4.2. Effects on Cellular Senescence and Cancer

Hyperinsulinemia associated with IR has many other negative actions. Among these, the action on cellular senescence and the development of tumors should not be overlooked. Senescent cells are characterized by the fact that they stop dividing and undergo specific changes, both in their appearance and activity. They produce specific molecules, contributing to the aging state of the entire organism. Cellular senescence is due to several factors, including oxidative stress and the presence of DNA damage. In particular, replicative senescence is linked to the so-called telomere attrition, a process that leads to chromosomal instability and promotes the onset of tumors [39].

It should be noted that increased cellular senescence is present in adult subjects with obesity, type 2 diabetes, and non-alcoholic fatty liver, regardless of age. In particular, Hyperin/IR stimulates cellular senescence in metabolic target organs such as the adipose tissue, muscle, liver, and brain in humans [Figure 3].

Among various published studies, one that was carried out on cultured human hepatocytes under chronic hyperinsulinemia and in knockout mice for insulin receptors in the liver (LIRKO mice) demonstrated a direct relationship between hyperinsulinemia and the senescence of hepatocytes, which was very interesting. This study also demonstrated that the dangerous effects of chronic Hyperin on the cellular senescence of hepatocytes can be blocked by reducing the number of insulin receptors or by the senolytic substances desatinib and quercetin [40,41,42].

Age predisposes to the development of many types of cancer; in fact, the incidence of numerous tumors increases with age, even if the underlying relationship has not yet been fully clarified. However, there is growing scientific demonstration that the increase in senescent cells in our body contributes to the advancement of tumors [43,44,45]. Recently, it has been shown that senescence acts as a tumor promoter and stimulates skin cancer by upregulating p38MAPK and MAPK/ERK signaling, upon which the high insulin levels associated with IR also act [46]. Insulin binds not only to its own receptors but also to those of insulin-like growth factor-1 (IGF-1), thus acting as a growth factor with pathologic consequences on the development of tumors [47,48].

In addition, there is a stimulating effect of Hyperin on the production of angiopoietin-2 (ANG-2) and therefore on angiogenesis which, in turn, is a mechanism necessary for tumor growth. In fact, a recent study has shown that serum levels of ANG-2 are higher in hyperinsulinemic subjects, and that insulin increases its expression and release by endothelial cells in vitro. Insulin induced the activation of p38 MAPK and cFOS. Furthermore, hyperinsulinemic plasma caused endothelial inflammation, which was reversed by an ANG-2-blocking antibody [49].

The careful observation of patients with cancer has established that Hyperin is a factor of great importance influencing the development of obesity, type 2 diabetes, and cancer. Both obesity and diabetes are considered risk factors for tumor onset and progression and the formation of metastases in many types of cancer. Furthermore, cancers are associated with difficult healing, a greater number of relapses, and greater mortality in patients with obesity or diabetes [50].

Patients with diabetes and/or subjects with metabolic syndrome have a doubled risk of developing cancer and cancer-related deaths [51,52,53]. Furthermore, it has also been reported in non-diabetic and non-obese subjects that Hyperin itself is related to increased cancer deaths, and therefore the authors underline that a treatment to reduce circulating insulin levels could be an important therapeutic approach for the prevention of cancer [48,53].

### 4.3. Effects on Brain

Another problem that should not be underestimated regarding the damage caused by hyperinsulinemia associated with IR is the very close relationship that exists between it and neurodegenerative diseases, although the mechanisms that link the two pathologies are not yet fully clarified. It has long been known that diabetes is associated with neurodegenerative diseases, probably because chronic hyperglycemia forms advanced glycation end-products, which in turn are able to modify key proteins such as amyloid β, tau, α-synuclein, and prions [54]. However, hyperinsulinemia may also play a role in itself, which would be of greater importance as an early phenomenon to be identified and corrected.

There are numerous pathophysiological hypotheses, partly well supported by the scientific literature, which try to explain the intricate mechanisms that associate Hyperin/IR with brain damage [55,56,57].

As regards the relationship between insulin and the brain, it must be remembered that, until a few years ago, it was believed that the brain was an organ insensitive to insulin actions. Instead, over the last 20 years, a considerable amount of the scientific literature has accumulated, which demonstrates that insulin penetrates the brain crossing the blood–brain barrier, where it binds to its specific receptors and regulates some important functions of the central nervous system such as the stimulation of appetite, cognitive behavior, and depression. It also controls some important systemic functions such as the production of glucose by the liver, lipogenesis and lipolysis, and the response of the sympathetic system to episodes of hypoglycemia [58]. Insulin binds to its receptors localized in different regions of the brain and initiates a series of phosphorylation reactions using two different receptor substrates (IRS-1 and -2) which, in turn, activate subsequent metabolic pathways. PI3K and protein kinase B (Akt) are kinases activated at the post-receptor level when insulin binds to its receptors and, in this way, acts on neuronal plasticity, survival, and neurotransmitter trafficking. Insulin also activates MAPK, which controls cell growth and proliferation. It should be underlined that the hippocampus, which is involved in cognitive function, and the hypothalamus, which controls peripheral metabolism, are characteristically rich in insulin receptors [59,60,61,62].

There are numerous epidemiological studies that highlight how the prevalence of Hyperin/IR is very high in patients with cognitive deficits or Alzheimer’s disease, reaching and exceeding the value of 81% of cases overall. In further detail regarding the close relationship between cognitive deficits, AD, and type 2 diabetes mellitus, it has been seen that these pathologies have multiple risk factors, comorbidities, and hypothetical pathophysiological mechanisms in common, so much so that, provocatively, it has been proposed to call AD type 3 diabetes mellitus [63,64,65]. Studies based on post-mortem examinations of the brains of subjects with cognitive deficits or Down syndrome have highlighted clear signs of IR in them, such as a significant reduction in receptors for insulin in the hippocampus, cortex, and hypothalamus [66].

Furthermore, a very interesting study carried out in Vervet monkeys showed that, shifting from a state of health to a stage of prediabetes and, subsequently, to frank diabetes, cerebral metabolism is altered with an increase in glucose and a decrease in amino acids and acylcarnitine. This alteration of metabolism stimulates the production and aggregation of amyloid β in a similar way in type 2 diabetes and AD, clarifying how some mechanisms present in diabetes can lead to cognitive deficits and dementia [67].

### 4.4. Effects on the Liver and Ovaries

There are important connections between liver function, Hyperin, and IR. In fact, as we have already seen, the liver is a very relevant organ from a metabolic point of view and has a fundamental role in the regulation of glucose and lipid metabolism. Impaired liver function leads to impaired hepatic metabolism that promotes the development of IR, which, in turn, is a common feature of both the development of non-alcoholic fatty liver disease (NAFLD) and the development of type 2 diabetes. There is a complex bidirectional relationship between NAFLD and diabetes. In particular, patients with NAFLD and IR have a high risk of developing diabetes early, while the majority of patients with type 2 diabetes more easily develop non-alcoholic fatty liver, non-alcoholic steatohepatitis, and other serious liver complications such as cirrhosis and hepatocellular carcinoma [68,69].

PCOS is a syndrome that has important metabolic and gynecological implications, including menstrual and conception difficulties. Affected women have a marked IR, which is independent of obesity, of a genetic nature, classified among the forms of type A IR. This form of IR in women is mostly recognized in adolescence due to the delay and alterations of the menstrual cycle. In women with IR and PCOS, insulin acts as a co-gonadotropin via a receptor that modulates ovarian steroidogenesis. Furthermore, the genetic alteration of insulin signaling in the brain contributes to impaired ovulation. For this reason, it has been suggested to treat this syndrome with insulin-sensitizing substances [70,71].

## 5. Possibilities of Treatment

There are many treatment options available once we identify the subjects with IR/Hyperin. There is no need to wait because waiting may result in the patient developing diabetes. In fact, in most cases, Hyperin associated with IR precedes the development of the described events by several years, even up to 15, and this can occur even in children [3,4,23]. For this reason, we must use a broad-based strategy, educating and acting effectively. We must educate the general population, even children, toward a more healthy lifestyle, with a balanced and more adequate caloric intake, accompanied by moderate, but constant over time, physical activity (walking at least 4000/6000 steps per day). It is not our intention to discuss in depth in this review the mechanisms by which physical exercise improves insulin sensitivity, but it is known that muscle training determines numerous beneficial adaptations in the affected skeletal muscles, including an increase in GLUT-4 expression [72].

Although it has not been perfectly clarified what the best amount and type of muscular exercise is to reduce IR, it has been verified that radical lifestyle changes lead to a long-term improvement in insulin sensitivity and reduce the incidence of overt type 2 diabetes in subjects with impaired glucose tolerance [73].

Unfortunately, this type of intervention may, in many cases, not be sufficient for various reasons, including the fact that many people are unable to constantly correct their lifestyle. Therefore, at this point, we are forced to intervene by helping the patient with the administration of insulin-sensitizing substances. In previous works we have reviewed many effective substances to choose from, both drugs and natural substances, depending on the tastes and characteristics of the patient [74,75,76,77].

Back in 2012, we provocatively published a manuscript entitled “Insulin resistance: Is it time for primary prevention?”, in the hope of inducing medical societies and national health authorities to take the problem into consideration [78]. Although little has been done, and precisely because of that, we still believe that the importance of the topic requires greater awareness on the part of the scientific and medical world. Scientific societies discuss the assessment of residual risk in subjects in whom the classic risk parameters appear to have been brought back to target. The concept of lowering total cholesterol and LDL cholesterol values in high-risk subjects is therefore further stressed.

Furthermore, the impact of the risk linked to chronic inflammation is being evaluated by studying markers such as C-reactive protein and interleukin-6, and the intervention on it with anti-inflammatory drugs. This begs the question of why neglect a risk factor such as IR with associated hyperinsulinemia, given that, among other things, it has been demonstrated that it itself causes chronic inflammation and produces significant damage over time? A large, controlled intervention study of IR associated with hyperinsulinemia is needed to test whether improving IR and reducing insulin levels can produce a prognostic improvement.

## 6. Conclusions

From the analysis of the available literature, it is clear that Hyperin/IR is an important risk factor for many pathologies. Therefore, with the aim of improving IR and reducing hyperinsulinemia, the early screening for IR in the general population should be recommended by medical societies and national health authorities in order to identify individuals affected by this condition and promptly begin preventive treatment. Lifestyle interventions through a balanced diet low in carbohydrates and constant physical exercise are well known preventive procedures. In case of insufficient compliance with those recommendations, the addition of insulin-sensitizing substances (drugs and/or food supplements) should be applied in the identified subjects in order to counteract the risk of diabetes, cardiovascular diseases, neoplasms, and neurodegenerative diseases.

## Figures and Tables

**Figure 1 biomedicines-12-02416-f001:**
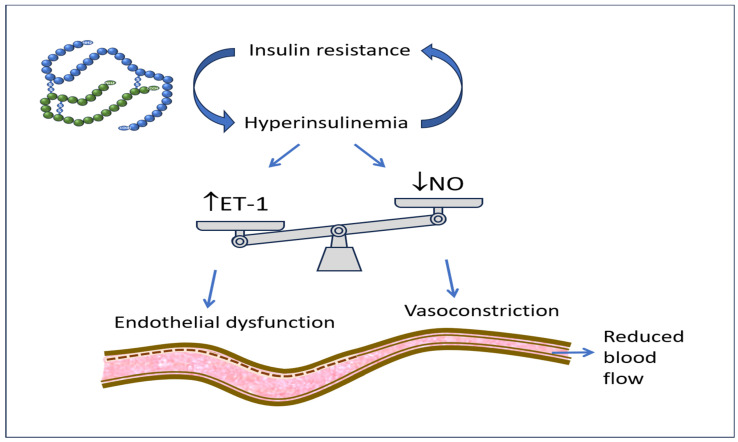
Insulin resistance with associated hyperinsulinemia increases ET-1 secretion and decreases NO availability, therefore producing vasoconstriction, reduced district blood flow, and endothelial dysfunction.

**Figure 2 biomedicines-12-02416-f002:**
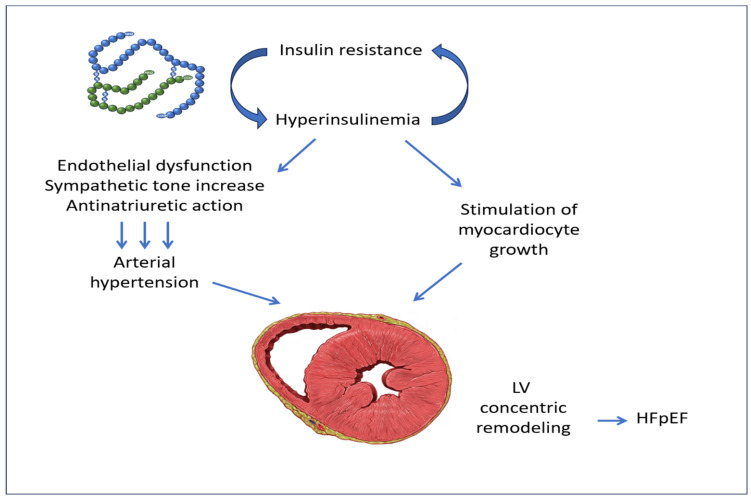
Insulin resistance with associated hyperinsulinemia, by producing endothelial dysfunction, increased sympathetic tone, and anti-natriuretic action, causes arterial hypertension, which, together with the stimulation of myocardiocyte growth by insulin, produces the pathologic concentric remodeling of LV. LV: left ventricle; HFpEF: heart failure with preserved ejection fraction. The image of LV concentric remodeling is from Patrick J. Lynch, medical illustrator, reproduced under Creative Commons Attribution 2.5 License 2006.

**Figure 3 biomedicines-12-02416-f003:**
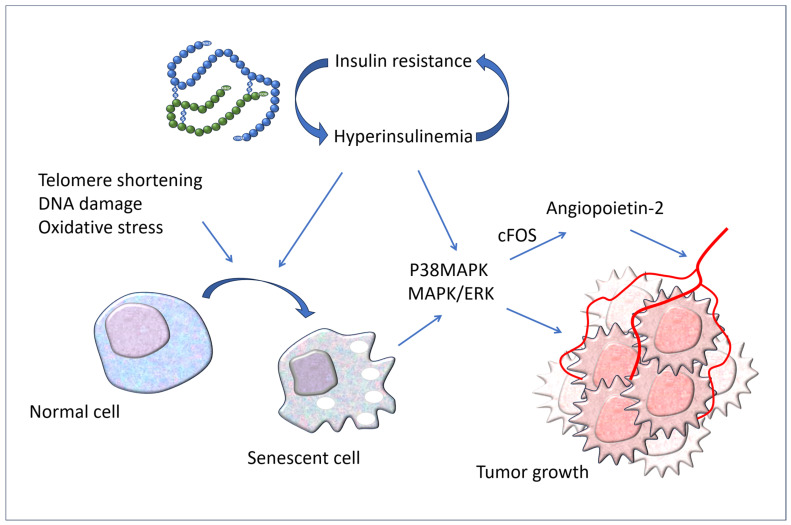
Hyperinsulinemia associated with IR causes cellular senescence in target organs of insulin actions. Both the increased circulating levels of insulin and cellular senescence stimulate cancer development by upregulating p38MAPK and MAPK/ERK signaling. P38MAPK: P38 mitogen-activated protein kinases; MAPK: mitogen-activated protein kinase; ERK: extracellular signal-regulated kinase.

**Table 1 biomedicines-12-02416-t001:** Indices of IR and their formulas.

IR Index	Index Acronym	Formula	Normal Values
**Homeostatic model assessment index**	**HOMA-IR**	Fasting blood glucose (mg/dL) × Fasting insulin (mU/L)/405	0.23–2.5
**Triglyceride–glucose index**	**TyG**	Ln [Fasting triglycerides (mg/dL) × Fasting blood glucose (mg/dL)]/2	<8.0
**Triglycerides/HDLc ratio**	**Ty/HDLc**	Fasting triglycerides (mg/dL)/Fasting HDL cholesterol (mg/dL)	<2.75 for men; <1.65 for women

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
