# Peer review of "Chronically Increased Levels of Circulating Insulin Secondary to Insulin Resistance: A Silent Killer"

_biomedicines, 2024, doi:10.3390/biomedicines12102416_

Round 1

Reviewer 1 Report (Previous Reviewer 1)

Comments and Suggestions for Authors

The authors focus on the classic internship between insulin, its pathological consequences, and the current therapeutic approaches in these cases. The authors provided an improved version of the manuscript (and they modified the title of the manuscript). I suggest expanding the inclusion and exclusion criteria used in their research. Please clarify – papers published in English, in extenso, abstract, period time of research.

Author Response

Rev. 1. I suggest expanding the inclusion and exclusion criteria used in their research. Please clarify – papers published in English, in extenso, abstract, period time of research.

Aut. We thank the Reviewer for his useful suggestion, and specified better the inclusion criteria used.

Reviewer 2 Report (New Reviewer)

Comments and Suggestions for Authors

I have completed the review of “Chronically Increased Levels of Circulating Insulin Secondary to Insulin Resistance: A Silent Killer" by Fazio et al. This review provides a thorough discussion of the importance of insulin resistance and its role in chronic diseases, such as cardiovascular and neurodegenerative disorders. Overall, it is an important contribution to understanding the problem of insulin resistance. However, I have some concerns, as detailed below:

-While the review covers various aspects of insulin resistance, it does not introduce significant novel findings. The manuscript could benefit from incorporating new insights or data that have not been extensively covered in the literature. For example, although the PI3K and MAPK pathways are discussed, they are already well-established in this field. The authors might consider addressing newer biomarkers or therapeutic targets. Including recent developments related to insulin resistance, such as the role of gut microbiota, immune modulation, or emerging diagnostic technologies, would make the review more compelling.

-The section on treatment discusses lifestyle interventions and insulin-sensitizing drugs, but it lacks a comprehensive comparison of different therapeutic strategies (e.g., diet, exercise, metformin, pioglitazone, SGLT-2 inhibitors, GLP-1 receptor agonists). A more detailed discussion of these strategies, particularly in terms of their comparative efficacy and potential applications for prevention, would enhance the clinical relevance of the paper.

Author Response

  1. Rev. While the review covers various aspects of insulin resistance, it does not introduce significant novel findings. The manuscript could benefit from incorporating new insights or data that have not been extensively covered in the literature. For example, although the PI3K and MAPK pathways are discussed, they are already well-established in this field. The authors might consider addressing newer biomarkers or therapeutic targets. Including recent developments related to insulin resistance, such as the role of gut microbiota, immune modulation, or emerging diagnostic technologies, would make the review more compelling.     1.Aut. We thank very much the Reviewer for his suggestion, and have added a brief paragraph on this issue.
  2. Rev.  The section on treatment discusses lifestyle interventions and insulin-sensitizing drugs, but it lacks a comprehensive comparison of different therapeutic strategies (e.g., diet, exercise, metformin, pioglitazone, SGLT-2 inhibitors, GLP-1 receptor agonists). A more detailed discussion of these strategies, particularly in terms of their comparative efficacy and potential applications for prevention, would enhance the clinical relevance of the paper. 2. Aut. Thank you for your observation. However, the aim of this review is to highlight the damage that can be caused by insulin resistance/hyperinsulinemia, if neglected, to try to stimulate scientific medical societies and national health authorities to take this issue into greater consideration. Therefore, in order to try not to make the manuscript boring and to mislead its purpose, we have not dealt with therapy, but have referred to our previous articles in which the therapy of insulin resistance was treated extensively. However, we have slightly modified the chapter according to your suggestion.

Reviewer 3 Report (New Reviewer)

Comments and Suggestions for Authors

The article Chronically increased levels of circulating insulin secondary to insulin resistance: A silent killer represent and interesting review article regarding a significance of chronically elevated insulin levels on numerous diseases development.

Article covers cardiovascular disease, malignant alteration, brain, liver and ovaries.

Authors could add more references, there are many novel research.

Regarding ovaries, please cite  the following article from 2024 and include adipose derived extracellular vesicles that are important in IR and PCOS. DOI: 10.1007/s12020-024-03702-w

In addition, authors could include more basic and translational research results with specific mentioning what was observed in animal, and what in clinical studies, and make some comparison. Authors could also provide one table to summarize major pathogenic mechanisms involved in chronic diseases development due to IR.

Conclusion has an important message regarding IR screening and one paragraph with future directions may be added before conclusion section to potentiate the importance of early development of IR and strategies for prevention of its complications

Author Response

Thank you for tour golf opinion on our manuscript.

  1. Rev. Authors could add more references, there are many novel research. 1. Aut. According to your suggestion, we have now added 18 references to the manuscript.
  2.  Rev. Regarding ovaries, please cite  the following article from 2024 and include adipose derived extracellular vesicles that are important in IR and PCOS. DOI: 10.1007/s12020-024-03702-w 2. Aut. According to your useful suggestion we cited this article. 
  3.  rev. In addition, authors could include more basic and translational research results with specific mentioning what was observed in animal, and what in clinical studies, and make some comparison. Authors could also provide one table to summarize major pathogenic mechanisms involved in chronic diseases development due to IR. 3. Aut. We have preferred not to add another table, but, according to your suggestion, we have discussed such mechanisms in the text. 
  4.  Rev. Conclusion has an important message regarding IR screening and one paragraph with future directions may be added before conclusion section to potentiate the importance of early development of IR and strategies for prevention of its complications. 4. Aut. Thanks for this helpful suggestion. According to this , we have added a paragraph before conclusions.

This manuscript is a resubmission of an earlier submission. The following is a list of the peer review reports and author responses from that submission.

Round 1

Reviewer 1 Report

Comments and Suggestions for Authors

The authors explore the classic relationship between insulin and its pathological consequences and the current therapeutic approaches in these cases. 

To improve the quality of the manuscript, I have the following suggestions for the authors:

- The authors should discuss the current knowledge regarding insulin resistance and its pathological context in the background.

- Although this is a review-type manuscript, extensive data are frequently associated with only one reference. Please see the 45-50 or 70-79 lines.

- Table 1 is not visible in the manuscript. Kindly organise the data from 107-114 lines in a table.

- To be more attractive to readers, kindly reconsider the structure and typographical features of all Figures.

- The authors used different keywords to explore this topic in some medical databases. The authors should clarify if they used inclusion and exclusion criteria during this part of the research. 

Literature data currently detailed the other etiological factors associated with insulin resistance, which are not discussed in the manuscript (e.g. nutritional imbalance, high-sodium diets, genetic factors, or different drugs – glucocorticoids, protease inhibitors or atypical antipsychotics).

The molecular mechanism regarding the potential role of insulin resistance in different pathological conditions is incompletely described (e.g. cardiovascular effects, cancer). Please see other studies in this area of research (Szablewski, L. 2024 doi.org/10.3390/ curroncol31020075; Ormazabal V, et al. 2018 doi.org/10.1186/s12933-018-0762-4; Kosmas C et al., 2023 doi: 10.1177/03000605231164548)

Kindly reorganise the reference list according to the Biomedicines journal recommendations.

Please revise the  English language in the main manuscript (please see 61-65, 81-83, or 136-137 lines).

Comments on the Quality of English Language

The authors should revise the  English language in the main manuscript (please see 61-65, 81-83, or 136-137 lines.

- In the presence of an IR condition, we have mostly a malfunction at the PI3K post receptor level which mainly regulates the metabolic actions and the formation of NO,  while the functioning of MAPK is little or not altered at all, so that the non-metabolic  actions of insulin and, in particular, the stimulating action on cell proliferation and on the secretion of ET-1 are exalted due to the chronic effects of Hyperin.

Although it is not yet perfectly clear whether the chicken or the egg came first, i.e.  whether the IR appears first and then the hyperinsulinemia (as one might think), these  two conditions are, in the vast majority of cases, chronically associated, and they are rapidly and constantly growing throughout the world.

In addition to this, which is already very important in itself, in IR conditions there is solid evidence of the fact that Hyperin is an important cause of arterial hypertension.

Author Response

 We thank very much the Reviewer for his useful suggestions.

The authors explore the classic relationship between insulin and its pathological consequences and the current therapeutic approaches in these cases. 

To improve the quality of the manuscript, I have the following suggestions for the authors:

  1. Rev. The authors should discuss the current knowledge regarding insulin resistance and its pathological context in the background. 1. Aut. We have added some sentences in the test on this.
  2. Rev. Although this is a review-type manuscript, extensive data are frequently associated with only one reference. Please see the 45-50 or 70-79 lines. 2. Aut. According to the Reviewer's suggestion we have added other references.
  3. Rev. Table 1 is not visible in the manuscript. Kindly organise the data from 107-114 lines in a table. 3. Aut.  We agree with the Reviewer and redid the table.
  4. Rev. To be more attractive to readers, kindly reconsider the structure and typographical features of all Figures. 4. Aut. We have completely modified the structure and typographical features of all figures. 
  5. Rev. The authors used different keywords to explore this topic in some medical databases. The authors should clarify if they used inclusion and exclusion criteria during this part of the research. 5. We have added a sentence to clarify this.
  6. Rev. Literature data currently detailed the other etiological factors associated with insulin resistance, which are not discussed in the manuscript (e.g. nutritional imbalance, high-sodium diets, genetic factors, or different drugs – glucocorticoids, protease inhibitors or atypical antipsychotics). 6. Aut. According to this Reviewer's suggestion we have added a short paragraph with references on this topic.
  7. Rev. The molecular mechanism regarding the potential role of insulin resistance in different pathological conditions is incompletely described (e.g. cardiovascular effects, cancer). Please see other studies in this area of research (Szablewski, L. 2024 doi.org/10.3390/ curroncol31020075; Ormazabal V, et al. 2018 doi.org/10.1186/s12933-018-0762-4; Kosmas C et al., 2023 doi: 10.1177/03000605231164548). 7. Aut. We tried to improve these paragraphs, also adding the 3 references suggested by the Reviewer.
  8. Rev. Kindly reorganise the reference list according to the Biomedicines journal recommendations. 8. Aut. Made.
  9. Rev. Please revise the  English language in the main manuscript (please see 61-65, 81-83, or 136-137 lines). 9. Thank you for this observation. We have submitted the test to a native English speaking.

Reviewer 2 Report

Comments and Suggestions for Authors

The current manuscript by Fazio and Affuso provides a comprehensive review of the hyperinsulinemia associated insulin resistance. The manuscript is well written and organized. It places emphasis on the complications of Hyperin/IR with a good summary of available treatment options. However, there are several points needing to be addressed.

1.       The table and figures are clear but roughly prepared. They can be revised to align with the typical format.  

2.       The literatures related to the GLP-1 receptor agonist are not up to date. There were several major updates on the clinical study such as the STEP-HEpEF trials, which should be included. PMID: 39226070, 39226070, 39226070, 39226070.

3.       There are some language errors that need to be revised, such as in line 144: Arterial hypertension resulted associated with …

4.       Line 56-60: “Among these…at the vascular level”. These statements are not accurate and misleading as it lacks the tissue/cell context.

Comments on the Quality of English Language

None

Author Response

We thank very much the Reviewer for his useful suggestions, which have permitted to improve the manuscript.

The current manuscript by Fazio and Affuso provides a comprehensive review of the hyperinsulinemia associated insulin resistance. The manuscript is well written and organized. It places emphasis on the complications of Hyperin/IR with a good summary of available treatment options. However, there are several points needing to be addressed.

  1. Rev. The table and figures are clear but roughly prepared. They can be revised to align with the typical format. 1. Aut. We agree with the Reviewer and have completely revised the table and figures.
  2. Rev. The literatures related to the GLP-1 receptor agonist are not up to date. There were several major updates on the clinical study such as the STEP-HEpEF trials, which should be included. PMID: 39226070, 39226070, 39226070, 39226070. 2. Aut. We removed the chapter on treatment to avoid burdening the manuscript and because the treatment of IR was not within the scope of the review. 
  3. Rev. There are some language errors that need to be revised, such as in line 144: Arterial hypertension resulted associated with … 3. Aut. Thank you, made.
  4. Rev. Line 56-60: “Among these…at the vascular level”. These statements are not accurate and misleading as it lacks the tissue/cell context. 4. Aut. The sentence was completely changed.